# Restoring Epigenetic Reprogramming with Diet and Exercise to Improve Health-Related Metabolic Diseases

**DOI:** 10.3390/biom13020318

**Published:** 2023-02-07

**Authors:** Merlin Jesso Abraham, Adham El Sherbini, Mohammad El-Diasty, Sussan Askari, Myron R. Szewczuk

**Affiliations:** 1Department of Biomedical and Molecular Sciences, Queen’s University, Kingston, ON K7L 3N6, Canada; 2Department of Surgery, Kingston Health Science Centre, Queen’s University, Kingston, ON K7L 2V7, Canada; 3Department of Physical Medicine and Rehabilitation, Queen’s University, Kingston, ON K7L 4X3, Canada

**Keywords:** diet, physical activity, health-related diseases, cardiovascular diseases, metabolic disorders, epigenome, epigenetic reprogramming

## Abstract

Epigenetic reprogramming predicts the long-term functional health effects of health-related metabolic disease. This epigenetic reprogramming is activated by exogenous or endogenous insults, leading to altered healthy and different disease states. The epigenetic and environmental changes involve a roadmap of epigenetic networking, such as dietary components and exercise on epigenetic imprinting and restoring epigenome patterns laid down during embryonic development, which are paramount to establishing youthful cell type and health. Nutrition and exercise are among the most well-known environmental epigenetic factors influencing the proper developmental and functional lifestyle, with potential beneficial or detrimental effects on health status. The diet and exercise strategies applied from conception could represent an innovative epigenetic target for preventing and treating human diseases. Here, we describe the potential role of diet and exercise as therapeutic epigenetic strategies for health and diseases, highlighting putative future perspectives in this field.

## 1. Introduction

Different patterns of lifestyle and behavioral factors, such as diet types and physical activity levels, have been associated with multiple health-related outcomes. The optimum balanced diet and regular exercise are the first lines for preventing chronic non-communicable illnesses, including cardiovascular diseases, metabolic disorders like type II diabetes mellitus (T2DM), and obesity as well as cancer [1]. A sedentary lifestyle and unhealthy diets, two significant contributors to global health morbidities and mortality, are also heavily dependent on personal choices, and social and physical environments. Chronic diseases are associated with reduced insulin sensitivity, chronic inflammatory status, gene dysregulation and dyslipidemias [2,3]. These combined factors are estimated to cause about 74% of deaths related to non-communicable diseases worldwide, accounting for about 41 million deaths yearly [4]. Although many studies have demonstrated the positive impact of pursuing healthy behaviors on reducing disease risks, the combined effects of both dietary and physical activity components that drive the biological mechanisms at the molecular level through epigenetic modifications have not been widely studied.

Epigenetics does not involve alterations in the DNA sequence but can control gene activity. A genome is the set of genetic information in the cell’s DNA. Every body cell has the same genome, though their respective phenotypes are different due to the control of gene expression [5]. An epigenome consists of a series of chemical compounds and complex external modifications that can cause changes in the genomic DNA without causing any changes to the DNA sequences of a gene, giving each cell a unique cellular and developmental identity [6]. The epigenome of an organism is dynamic and flexible and can influence its phenotype by turning the genes on and off, based on the environment to which it is exposed [5]. Some significant epigenetic changes detected are DNA methylation, histone modifications and RNA-associated alterations. These modifications are potentially heritable across the genome, leading to transgenerational inheritance [7]. They can even be reversible, based on lifestyle modifications [5]. Therefore, this review focuses on the impact of environmental factors, including diet type and exercise, and the subsequent combined effects on epigenetic modifications, such as DNA methylation, that may aid in preventing health-related disorders.

## 2. Impact of Diets on Metabolic Diseases through Epigenetic Modifications

Nutrition is one of the most well-studied environmental epigenetic factors. Several studies have reported the epigenetic effects of diet on the phenotype and disease susceptibility of individuals throughout their life [8]. Prenatal nutrients supplied with methyl-donating foods, such as choline and folate, are essential for fetal development in the early stages of pregnancy. This diet is required for DNA methylation status and the resultant impact of gene expression [9]. Then, the epigenetic mechanism of DNA methylation involves adding a methyl residue at the position 5 of the pyrimidine ring of cytosine. Protein transcription occurs at promoters of the DNA that are abundant with cytosine-phosphate-guanine (CpG) sites. Beside them lies the structure of chromatin, which generally promotes transcription based on whether the CpG sites are methylated [5]. Recent reports have presented that non-CpG methylation is upregulated in differentiated and somatic tissues, and the low detection is caused by technical biases [10]. If position 5 of the CpG island is methylated, it represents an inactive promoter, resulting in chromatin condensation by imparting tight compaction to the structure, preventing transcription [5]. Thus, DNA methylation plays a significant role in tissue-specific gene regulation and transcription throughout life. It has been suggested that DNA methylation is a natural link between genetic susceptibility and environmental exposures in common diseases [11]. This methylation process is facilitated by the enzymes DNA methyltransferases (DNMTs) [5]. In recent years, reports have shown that a significant fraction of DNA methylation sites positively correlate with gene expression [12], challenging the traditional view that DNA methylation represses gene expression. However, Wan et al. [13] demonstrated that the expression of genes’ negatively correlated tissue-specific differentially methylated regions (T-DMRs) are enriched for functions carried out in adult tissues. In contrast, the positively correlated genes are enriched for negative regulators such as transcriptional repressors. Interestingly, this two-tier mechanism regulated by positive T-DMRs may be specific to the development and the establishment of tissue-specific expression. In addition, gene regulation can be modulated depending on the location of methylation in the transcriptional site [14].

If positive T-DMRs are specific to development [13], the question is whether epigenetics and epigenetic inheritance influence evolution [15]. Trans-generational epigenetic inheritance must be considered when estimating quantitative genetic parameters, but it can also respond to selection and influence adaptation to new environments [13]. For example, the transgenerational effects of poor maternal diet during pregnancy can lead to health disorders and disease susceptibility or cancer throughout the offspring’s life [16,17]. For instance, studies investigating the impact of starvation during the Dutch Famine in 1944–1945 during pregnancy found that the children developed subsequent adverse health and developmental outcomes such as the increased risk of T2DM, cardiovascular diseases, metabolic disorders and other cognitive dysfunctions compared with control counterparts [9,16]. It is important to note that paternal dietary practices were also shown to transmit health complications in offspring through transgenerational epigenetic modulation of the spermatozoon nucleus in many historical cohorts. In a study conducted in a remote Swedish town, restrictive access to nutritional foods in one generation was associated with mortality rates two generations later in a gender-specific manner, i.e., paternal grandfather’s dietary supply was related to grandsons’ altered metabolism, and paternal grandmother’s dietary supply was linked to granddaughters’ altered metabolism [18]. Another study suggested that the grand offspring of males, who were exposed to famine while in utero, suffered from higher BMIs than the control population [9]. The transmission of paternal and maternal diet patterns influencing nutritional diets in early life development can induce permanent changes in DNA methylation, impacting age-related diseases and health throughout the individual’s life span [19]. Nutrients can inhibit directly the DNA methyltransferases (DNMTs) that are required for DNA methylation or indirectly by regulating the substrates available for the enzymatic mechanisms [20].

It was suggested that fetuses receiving a limited supply of nutrients in the in utero environment epigenetically could adapt to that environment. They tended to have low birth weight and were smaller for gestational age [21,22]. However, it was found that, if these subjects were exposed to accelerated growth utilizing a high-calorie nutritional environment later in their childhood, they developed a ‘mismatch’ between the postnatal nutritional environment and the environment predicted by nutrition scarcity in pregnancy. This disruption in the energy balance can trigger epigenetic changes, predisposing these individuals to insulin resistance and obesity and a higher risk of developing chronic metabolic illnesses later in life [23,24].

### 2.1. Impact of Dietary Patterns on Metabolic Diseases such as T2DM through Epigenetic Modifications

Conversely, dietary restrictions with good nutritional content have been shown in several models to extend lifespan. In cases of adverse environmental exposures, studies have shown that their effects can be counteracted by intaking a nutrition-rich diet of fruits, vegetables, legumes and high-quality whole foods [9]. The plant-based or ‘vegan’ diet generally contains whole grains, legumes, vegetables, fruits and nuts and abstention from meat and dairy products. Plant metabolites containing polyphenols can inhibit DNA methyltransferase (DNMT) involved in preventing cancer, metabolic diseases and other chronic illnesses. Contrarily, the vegan diet has also been associated with vegan-associated B12 deficiency, possibly leading to diseased aging and hypomethylation [25]. DNA methylation, catalyzed by DNMTs, is essential in maintaining genome stability (Figure 1). Many types of cancer are associated closely with aberrant expression of DNMT involving DNA methylation pattern disruptions [26]. Recent Adventist Health cohort studies conducted on comparative DNA methylation analyses of an epigenome-wide approach from the blood cells of vegans identified numerous CpG sites that displayed differential DNA methylation [27]. Most of these differential methylations of genes detected were hypomethylated in vegans, and the most significant differences were found in the gene bodies. Theoretically, the substantially higher levels of bioactive compounds and phytochemicals such as isoflavones, carotenoids and omega-3 fatty acids in plasma, urine and adipose tissue can also modulate methylation [27]. Similar observational studies evaluated the relationship between diet types and incidence of T2DM in participants of different ethnicities, and cases of diabetes developed in 0.54% of vegans compared with 2.12% of non-vegetarians after a two-year study [28]. Vegan diets are also associated with reduced body weight, increased insulin sensitivity, and improved glycemic and lipid control, thus being recommended for lower diabetes risk [28]. Clinical trials conducted in individuals with T2DM demonstrated that a vegan diet significantly improved glycemic control over conventional diabetic diets, thus presenting potential advantages for managing the disease [29].

### 2.2. Impact of Dietary Patterns on Cancer Prevention through Epigenetic Modifications

Dietary compounds to prevent cancer development through epigenetic mechanisms are another rapidly evolving concept. A substantial amount of evidence indicates the role of atypical DNA methylation as a trademark for many types of cancer development. General DNA hypomethylation occurs in cancer cells commonly, leading to chromosomal instability and genetic mutations. However, gene-specific promoter hypermethylation of tumor suppressor genes in cancers leads to their silencing [31,32]. As such, epigenetic biomarkers are utilized for cancer risk prediction, diagnosis, treatment and response. Therefore, they can provide insightful etiologic information about how the epigenetic modifications can be reversed for specific targets in chemoprevention [32].

Interestingly, many bioactive compounds in dietary agents mediate favorable epigenetic changes for cancer prevention and chemotherapeutic strategies. For instance, the anti-cancer properties of (−)-epigallocatechin-3-gallate (EGCG), polyphenolic catechin found in green tea, have been demonstrated to inhibit numerous cancer types by directly restricting DNMT activity [31] that can lead to potentially reversing the epigenetic silencing of tumor suppressors such as *CDKN2A* and *O*^6^-methylguanine-DNA methyltransferase by cancer cells [32]. Quercetin is a polyphenolic compound found in red wine, apples and onions. This Quercetin flavonoid can block the cell-cycle process in only cancer cells by inducing pro-apoptotic effects [33]. When given with curcumin, Quercetin decreased DNMT levels, leading to global hypomethylation and induction of apoptosis through mitochondrial depolarization. These effects consequently revived androgen receptors in prostate cancer cells lacking androgen receptors [33]. It is also important to note that, to elicit these positive epigenetic alterations of chemoprevention, the bioactive metabolite in the diet must enter circulation at adequate concentrations to reach the target tissue. Therefore, diet intake, dosing and bioavailability determine the effectiveness of the dietary compound of interest [32,33]. As everyday diets only includes small concentrations of different bioactive compounds, it is essential to consider dietary-based epigenetic cancer prevention starting as an early-life intervention and extending across the lifespan for continued efficacy [32].

### 2.3. B Vitamins, One-Carbon Metabolic Molecules, Biochemical Conduits Interlinking DNA Methylation, Associated Proteins and RNA

Metabolism involving one-carbon dietary constituents such as folate (B9), B vitamins (B2, B6, and B12), methionine and choline (betaine) are essential substrates or cofactors [34]. These authors reported that inadequacies of these dietary constituents, contributing to classical dietary deficiency syndromes, can also cause neural tube defects, cardiovascular disease and cancer. Clare et al. [35] have proposed that a one-carbon B vitamin metabolism is a functional biochemical conduit regulating early offspring development involving the parental environment and epigenetic. Interestingly, interindividual and ethnic variability in epigenetic-gene regulation may arise due to genetic variants within the one-carbon genes, associated epigenetic regulators, and differentially methylated target DNA sequences [35]. Mentch et al. [36] suggested that an in-depth specific mechanistic analyses of one-carbon metabolism and epigenetics might provide an understanding of the relations between the effect on gene expression, whether transient or permanent, and the status of metabolic activity and histone methylation. Here, the findings might also provide a molecular basis for a nutritional diet influencing gene expression via cell metabolism.

Lyon et al. [37] reported that most B vitamins are involved in one-carbon metabolism either directly or tangentially. This one-carbon metabolism can generate methyl donors by S-adenosylmethionine (SAM), which are utilized by RNA, DNA, histones and protein methyltransferases. Here, the methylation process is important for cellular metabolism, such as epigenetic regulation and protein–protein interactions, involved in embryonic development, cognitive function and hematopoiesis [38,39]. In addition, any perturbations in the B vitamins’ uptake and homeostasis, causing intermediates deficiencies or even an excess of one-carbon metabolism, can lead to anemia, neurological defects, cancer and defective immune responses [7,8,9]. The central role of vitamin B9 (folate) and vitamin B12 (B12) in human health is directly involving the one-carbon metabolism process [37].

Raghubeer et al. [40] reported on some of the one-carbon metabolic cycle functions, including the methionine metabolism of folate, DNA, RNA and protein synthesis. The 5-10-methylenetetrahydrofolate reductase (MTHFR) enzyme maintains the balance of methionine and homocysteine (Hcy), preventing cellular dysfunction. Polymorphisms in the C677T MTHFR gene can contribute to various diseases, such as cardiovascular diseases, cancer, inflammatory conditions, diabetes and vascular disorders. The C677T MTHFR polymorphism is thought to be the most common cause of elevated homocysteine levels, considered an independent risk factor for cardiovascular diseases.

Mahmoud and Ali [20] reported on the possible highlights of the “one-carbon metabolism” cycle on the levels of nutrient exposure impacting DNA methylation in the human body (Figure 2). In their example, nutrients either maintain or change the balance between s-adenosylmethionine and s-adenosylhomocysteine, regulating the availability of methyl donors and the activity of DNMTs.

## 3. Impact of Exercises on Metabolic Diseases through Epigenetic Modifications

Regular physical activity is another prominent lifestyle factor that protects against many chronic diseases [41]. Physical exercise and activities are often used interchangeably, and it is essential to differentiate between them. The former refers to energy expenditure from any body movement more than resting. At the same time, the subcategory of physical training, like exercise activity, is more organized and structured, and thereby aiming to improve and maintain physical fitness. Typically, it involves contracting voluntary muscles and body movements to alleviate symptoms, improve functions and delay health decline. Exercise training can improve an individual’s health and well-being [42]. Consistent evidence demonstrates exercise’s physiological and metabolic adaptations, mediated by, or which may result in, epigenetic modifications on various muscle tissue types.

### 3.1. Exercise-Induced Epigenetic Modifications in Genes

Overall health of an individual can be improved by regular physical activity. Exercise can influence epigenetic modifications [42]. Diet [43] and physical exercise [42], or both, can alter the epigenome. Barrón-Cabrera et al. reported an excellent systemic review of exercise interventions affecting epigenetic modifications [44]. Interestingly, exercise-induced gene modifications have been known to affect insulin resistance and type-2 diabetes, obesity, inflammation, and cardiovascular disease and blood lipid alterations. The studies indicate that exercise can alter the epigenome and that these outcomes could affect specific metabolic pathways (Figure 3). However, the studies reported to date may have several limitations, including sample sizes, different ethnic populations, different exercise interventions, the available tests for exercise, and the varied epigenetic modifications analyzed in different tissues.

### 3.2. Crosstalk between TOLL-like and Insulin Receptors and G Protein-Coupled Receptor Signaling

A G protein-coupled receptor (GPCR) molecular signaling platform was reported by us that was essential for the activation of glycosylated receptors and its targeted translation in human disease [46,47]. This receptor signaling axis is depicted in Figure 4 and described in detail by Abdulkhalek et al. [46]. Here, specific ligand binding to its receptor induces an endogenous mammalian Neu1 and matrix metalloproteinase-9 (MMP9) crosstalk in activating the receptor. Here, Neu1 and MMP9 form a complex tethered at the ectodomain of glycosylated receptors such as TOLL-like and insulin receptors on the cell surface. This receptor signaling axis proposes that ligand binding to its receptor causes a conformational change of the receptor, which results in the activation of neuromedin B GPCR (NMBR), also tethered to the receptor. Activated NMBR initiates Gαi-protein signaling, which triggers the activation of MMP9 to induce Neu1 subsequently. Here, activated MMP9 removes the elastin-binding protein (EBP) as part of the molecular multi-enzymatic complex that contains Neu1 and protective protein cathepsin A (PPCA) [48]. Activated Neu1 specifically hydrolyzes the α-2,3-sialyl residues linked to β-galactosides of the receptors, which are distant from the ligand binding sites. This prerequisite process by activated Neu1 is to desialylate the receptors, removing glycosylation steric hindrance of receptors and facilitate receptor association. This signaling axis depicted in Figure 4 describes a GPCR-MMP9-Neu1 receptor signaling platform induced by receptor ligand binding with subsequent receptor activation on the cell surface. These unique receptor signaling interactions are proposed to control the fundamental downstream signaling mechanisms involved in human disease. Here, the initial induction of GPCR and Gai-signaling is an essential process in this paradigm, the process of which was previously reported by us [47] and others [49]. Notably, different GPCR agonists such as bombesin, bradykinin, lysophosphatidic acid (LPA), cholesterol, and angiotensin-1 and -2 can transactivate Neu1 through the intermediate MMP9 in order to induce the activation of TLRs [50] and insulin receptors [51] with subsequent cellular responses.

### 3.3. TLR-Induced Epigenetic Changes

Perkins et al. [52] reported a review of the status of the chromatin profile expressions of inflammatory genes in response to TLR stimulation and how this chromatin landscape affects cell type specific and temporal gene expression. Notably, TLR can induce epigenetic changes constituting both the positive and negative regulation of TLR-induced genes. However, signaling pathways and transcription factors play a pivotal role in chromatin-based mechanisms affecting proinflammatory responses and epigenetic marks in rewiring context-specific gene expressions in innate immune cell types.

Interestingly, TLRs are receptors for pathogen-associated molecular patterns (PAMPs) recognizing antigenic epitopes of microbial pathogens, or danger-associated molecular patterns (DAMPs), from stressed or injured cells. They can also be involved in the progression of cardiovascular diseases [53]. Myocardial and arterial cells have been observed to express TLRs. Mouse knockout and human polymorphism studies reveal TLR3 in neointima formation and atherosclerosis. Notably, TLRs may be associated with several cardiovascular diseases [54], such as in the pathogenesis of microvascular complications of diabetes, atherosclerosis, viral myocarditis, dilated cardiomyopathy, sepsis-induced left ventricular dysfunction and cardiac allograft rejection.

### 3.4. Crosstalk between Insulin Receptor and G Protein-Coupled Receptor Signaling

Obesity, insulin resistance and type 2 diabetes are usually referred to as a chronic low-grade metabolic meta-inflammation [55], and have been extensively reviewed [55,56]. Oh et al. [57] reported that omega-3 fatty acids binding to their GPR120 receptor can mediate potent anti-inflammatory and insulin-sensitizing effects. The GPCR GPR120 receptors bind long-chain fatty acids such as palmitoleic acid, the omega-3 fatty acids (α-linolenic acid, docosahexaenoic acid and eicosatetraenoic acid.

The bombesin-related peptides bind closely related GPCR receptors in mammals, referred to GPCRs neuromedin B receptor (NMBR), a GRP receptor (GRPR), and an orphan bombesin-receptor subtype-3 (BRS-3) [58]. Using NMBR knockout (KO) mice, partial resistance to diet-induced obesity was observed [59]. However, the disruption of the NMBR signaling pathway in these KO mice did not affect the body weight or food intake (fed a normolipid diet). Notably, they developed partial resistance to diet-induced obesity.

The GPCR neuromedin B (NMBR) receptor was reported to form a tethered complex with TLR4 [50] but also with TLR-7 and -9 [60], epidermal growth factor receptor (EGFR) [47] and insulin IRβ receptor [61]. In the report on intracellular TLRs, the NMBR co-immunoprecipitated (co-IP) with MMP9, and conversely, MMP9 co-IP with NMBR isolated in macrophage cell lysates from naive and TLR9 ligand stimulated cells [60].

### 3.5. Proposed Cellular and Biochemical Changes Causing Epigenetic Changes

Kanherkar et al. [5] reviewed the effects of an epigenetic factor(s) that can be manifested as a global change in DNA methylation, affecting multiple genes or modified expression of particular genes. However, the mechanisms and cellular and biochemical pathways involved in creating these global or specific epigenetic changes are currently unknown. They proposed that epigenetic changes can act through either a direct or an indirect mechanism(s). For example, a direct epigenetic effect could occur in two ways. A type 1 epigenetic direct effect can occur when the epigenetic mark alters the epigenetic enzymes directly, either by binding to them and inhibiting them from carrying out their normal function, or altering or upregulating them. The altered bioavailability of epigenetic enzymes can result in the aberrant recruitment of epigenetic tags to specific promoters or enhancers on a genome-wide scale. Such a direct effect would be effective across the entire genome, resulting in a randomly altered epigenome but not affecting any specific gene. For example, a type 1 epigenetic factor is anti-hypertensive hydralazine, directly inhibiting DNA methylation. A type 2 epigenetic direct effect occurs when it causes a change in a biochemical process, resulting in an altered availability of a substrate, intermediate, by-product or any other metabolite participating in the biochemical pathway that is used to make up epigenetic tags (for example, acetate). This process, in turn, could lead to altered availability of epigenetic tags such as acetyl groups on histones, leading to non-specific modification of the epigenome. However, nutritional modifiers associated with epigenetics may potentially work through alterations of the regulation of biochemical pathways, associated with the epigenetic markers (e.g., B vitamins, modulation of one-carbon metabolism) [62]. Given that the sole genes physiologically are regulated by these pathways, the previously mentioned changes could result in gene-specific epigenetic modulation [5].

The indirect mechanisms of epigenetic changes are considered biphasic in nature. For example, an acute exposure to an environmental factor(s) can influence cellular signaling downstream pathways, altering growth factors, receptors and ion channels expressions. These factors as a result can alter transcription factor activity at the gene promoters. With a more chronic exposure altering gene expression activity, the transcription factors and other gene regulatory proteins can actually recruit or repel epigenetic enzymes to/from the associated chromatin, resulting in the addition or removal of epigenetic tags.

Tiffon [9] reviewed environmental epigenetics, describing how environmental factors affect cellular epigenetics and human health. Environmental epigenetic factors include behaviors, nutrition, chemicals and industrial pollutants. Epigenetic mechanisms are also implicated during development in utero and at the cellular level, so environmental exposures may harm the fetus by impairing the epigenome of the developing organism to modify disease risk later in life [9].

Some of the proposed environmental factors that can influence cellular signaling pathways are depicted in Figure 4: diet, exercise, diet-induced adaptive thermogenesis and gut microflora metabolism. Adaptive thermogenesis involves the regulation of body heat production in response to temperature and diet, which, in turn, can result in inefficiency of metabolic activity. For example, a critical regulator of diet-induced thermogenesis and bone homoeostasis is the neuropeptide FF receptor-2 (NPFFR2) [63]. A defective NPFFR2 signaling can lead to a reduction in brown adipose tissue (BAT) thermogenesis under high-fat diet (HFD) conditions, which can significantly reduce a transcriptional coactivator, peroxisome proliferator-activated receptor gamma coactivator-1 alpha (PGC-1α), affecting the biogenesis of mitochondrial metabolism, respiratory, oxidative phosphorylation, fatty acid β-oxidation and uncoupling protein-1 (UCP1) levels in the brown adipose tissue. NPFFR2 signaling can elevate thermogenesis from the diet through a novel circuitry of hypothalamic NPY-dependent coupling energy homeostasis, resulting in an energy partitioning to bone and adipose tissue [63]. An indirect mechanism of altering the epigenetic markers may involve a biased NPFFR2 GPCR transactivation of the Neu1-MMP9-GPCR signaling receptor platform in regulating several receptor tyrosine kinases (RTK) and TLRs influencing epigenetic rewiring contributing to the molecular targeting of multistage of human diseases (Figure 4). In addition, NPFF presents different responses in parvocellular neurons, which regulate sympathetic outflow and magnocellular neurons regulating secretion of neurohypophysis hormones of the paraventricular nucleus, all of which play a role in central regulation of blood pressure and cardiovascular disease [64]. Indeed, genome and nutrient interactions appear to have an essential role in disease prevention and health maintenance. Nutrigenomics and nutrigenetics address how epigenetics and genetics explain individual dietary susceptibility [65].

### 3.6. Exercise-Induced Hypertension Is Associated with Angiotensin II (Ang2) Activity

Angiotensin II (Ang2) can mediate exercise-induced hypertension, which adversely impacts future cardiovascular health. Kim et al. [66] evaluated the total nitric oxide activity and the renin–angiotensin–aldosterone system in middle-aged marathoners with exercise-induced hypertension. Ang2 levels were reduced following 30 min of exercise, whereas the renin level was elevated. No angiotensin I and angiotensin converted enzyme level (ACE) changes were observed. The authors concluded that reduced nitric oxide and Ang2 levels are associated with exercise-induced hypertension in middle-aged long-distance runners [66].

Interestingly, Jiang et al. [67] reported on the mortality of individuals already burdened with cardiovascular disease that epigenetic age acceleration remains strongly predictive of mortality. A member of the Ang family called angiopoietin-2 (Ang2), involved in developmental angiogenesis and in the growth of human cancers, may have mortality associations. Ang2 regulates endothelial permeability and angiogenic functions, suggesting that specific vascular pathophysiology may link accelerated epigenetic aging with increased mortality risks [67]. On the other hand, Ang1-7 is an endogenous peptide fragment produced from Ang1 or Ang2 via endo- or carboxy-peptidases, respectively. Ang1-7 can positively affect metabolism by increasing glucose uptake and lipolysis, while decreasing insulin resistance and dyslipidemia [68]. Ang1-7 can also improve cerebroprotection against ischemic stroke, besides its effects on learning and memory [69]. Ang2 binds to its BR2 GPCR and Ang1-7 via its Mas GPCR.

The crosstalk between insulin receptor and GPCR signaling is well recognized for the regulation of multiple physiological functions as well as the pathogenesis of important diseases, including cancer, obesity, metabolic syndrome, hypertension, type II diabetes mellitus, insulin resistance and hyperinsulinemia [70]. Ang-1-7 acting through Mas GPCR can enhance the insulin sensitivity of skeletal muscle after a bout of exercise [71].

G protein-coupled receptors (GPCRs) can participate in several signaling pathways, and this property led to the concept of biased GPCR agonism. Haxho et al. [49] reported a biased GPCR agonism as small diffusible molecules activating Neu1-mediated insulin receptor signaling. GPCR agonists, significantly bombesin, bradykinin, Ang1 and Ang2, dose-dependently induce Neu1 sialidase activity and insulin receptor activation without insulin. Figure 4 depicts the molecular link regulating the interaction and signaling mechanism between these molecules on the cell surface.

Due to the complexity and differences in the literature, a recommendation about the type, intensity or duration of exercise is currently not possible for varied subsets of the population, such as healthy, diseased and trained individuals.

Interestingly, immunotherapy and exercise can have a synergistic effect. Reports have indicated that exercise can mobilize immune cells, which results in their redistribution to different body compartments [44,72]. In preclinical models, exercise can lead to immunological changes in the tumor microenvironment [72]. Acute exercise is considered an essential advent for the immune system to stimulate the ongoing exchange of leukocytes between circulation and tissues. In contrast, exhaustive exercise and high-intensity athletic training have an adverse increased risk of illness [73].

Interestingly, exercise involving marathon athletics may involve an inflammatory profile and increased risk of injury. There is an in silico association between the soluble mediators of inflammatory and circulating-inflammatory miRNA profiles and the pathways of cancer, immune system disorders and inflammation processes [74]. These observations suggest an interconnection between stress induced by acute exercise and the activation of the immune system versus chronic exercise-induced stress.

### 3.7. Impact of Exercise on Epigenetics in Skeletal Tissues

Several studies have been conducted on skeletal muscles to understand the impact of exercise training through epigenetic rewiring. Skeletal muscle shows plasticity in adapting to stressor factors, affecting the metabolic and structural processes of the tissue. During physical exercises, muscle contraction can affect adaptive responses by enhancing the metabolic efficiency, oxidative capacity and contractile activity. Here, altering gene expression profiles and protein levels of genes are known to regulate mitochondrial function and fuel utilization through DNA methylation. In a methylation association study conducted on skeletal muscle in healthy individuals before and after acute exercises, Barrès et al. [75] found a decrease in global DNA methylation after an acute bout of exercise. This finding was further evaluated for gene specificity in skeletal muscles, especially for the ones suggested to be differentially methylated in T2DM [17], transcripts of which are elevated after exercises, such as peroxisome proliferator-activated receptor-gamma coactivator (*PGC-1α*), transcription factor A, mitochondrial (*TFAM)*, peroxisome proliferator-activated receptor delta (*PPAR-δ*), pyruvate dehydrogenase kinase 4 (*PDK_4_*), and citrate synthase (*CS*); all these transcription factors are involved in muscle-specific gene expression and housekeeping genes. Following methylated DNA immunoprecipitation and quantitative PCR, the results revealed that methylated promoters of metabolic genes were lower after acute exercise. In contrast, muscle-specific transcription factors such as myogenic differentiation 1 (*MYOD1*) and myocyte-specific enhancer factor 2A (*MEF_2_A*), as well as glyceraldehyde-3-phosphate dehydrogenase (*GAPDH*), were all unaffected [75].

This study analyzed the kinetic time-course process and concentration dose-response of DNA methylation after acute exercise in another cohort. Biopsies of skeletal muscle tissue were obtained before, immediately after, and 3 h after 40% peak aerobic capacity (low-intensity) or 80% peak aerobic capacity (high-intensity) acute exercise training. The high-intensity aerobic exercises measure the maximum oxygen levels the body can utilize during exercise. It was found that high-intensity exercise markedly led to DNA hypomethylation in the promoter regions of *PGC-1α, TFAM, MEF2A,* and *PDK4* following exercise. At the same time, *PPAR-δ* methylation was decreased 3 h after exercise. These effects suggest that DNA methylation is essential to the exercise-induced effect on many metabolic gene expressions. It can produce more significant responses at higher exercise intensities, suggesting that epigenetic modifications on genes are mediated in an exercise-intensity-dependent manner [75].

Another study found that, after a 6-month endurance training intervention in healthy subjects with a family history of T2DM, DNA methylation of genes including MEF*2A*, RUNX*1*, NDUFC*2* and *THADA* involved in retinol metabolism and calcium-signaling pathways were decreased after exercise in skeletal muscle biopsies with known functions in muscle and T2DM. These changes involved in exercise-induced regulation of *GLUT4* expression may potentially influence glucose uptake in muscle, thereby reducing the future risk of T2DM by modulating the mechanism epigenetically [76]. Another interesting finding is that regular exercise training could reduce inflammation and its associated complications, which may be beneficial for attenuating many chronic illnesses. For example, interval walking at 40% peak aerobic capacity, followed by 70% peak aerobic capacity in older men, stimulated hypermethylation in the *ASC* gene promoter. This finding was associated with lower IL-1β and IL-18 levels related to rheumatoid arthritis, atherosclerosis and T2DM.

However, there is an undesired effect on performing submaximal and prolonged exercises, such as the susceptibility to skeletal muscle damage, cardiac stress, necrosis and systemic inflammation. These alterations were found in marathon runners, and it was noted that their circulating inflammatory miRNA levels, such as miR-1, miR-133a and miR-206, were substantially elevated [44]. MicroRNAs (miRNAs) are small non-coding RNAs. They negatively regulate gene expression at the post-transcriptional level and are involved in critical biological processes, including development, homeostasis and aging. This also suggests the presence of myo-miRs as biomarkers of muscle damage, injured or stressed cells, and adaptation. Alternatively, another study observed upregulation of miR-1, miR-133b and miR-206 in healthy female subjects after 12 days of walking training at a low-to-moderate intensity at low altitudes. These miRNA levels were associated with mitochondrial apoptosis inhibition and these results were attributed to an inflammation-mediated skeletal muscle regenerative capacity in response to exercise. There was also an associated concomitant decrease in O_2_ consumption and increased intracellular calcium concentrations in the subjects [77]. In conclusion, the exercise-induced inflammatory response seems to depend on the exercise’s intensity, time and distance, determining the harmful effect on muscle tissues mediated by epigenetic mechanisms [44].

### 3.8. Impact of Exercise on Epigenetics in Cardiac Tissues

Exercise is recognized as one of the essential preventative strategies against the development of cardiovascular disorders; however, the molecular mechanisms relative to this association remain unclear [78]. Understanding covalent histone post-translational modifications and non-coding RNAs underlying epigenetic mechanisms may explain the cardioprotective effects of exercise [78]. Several studies have evaluated the association between exercise and epigenetic modifications on cardiac tissue [78].

It is well determined that exercise, precisely aerobic exercise, is well associated with an upregulation of cardiac angiogenesis [79]. Multiple studies have shown that exercise alters the exercise-induced expression of miRNAs. These studies have assessed the impact exercise and miRNA expression have on pro-hypertrophy, cardiac growth, anti-angiogenesis and antifibrotic functions of the cardiac muscle. Wardle et al. evaluated how circulating miRNA (c-miRNA) differs between endurance and strength athletes [80]. Fourteen c-miRNAs were investigated through the collection of plasma from both cohorts. It was found that miRNAs, such as miR-222, miR-21, miR-146a and miR-221, were upregulated in the plasma of endurance athletes compared with strength athletes.

Similarly, Baggish et al. [81] reported that c-miRNAs originating from cardiac tissue, miR-208a, were associated with aerobic exercise. These c-miRNAs were significantly upregulated following a marathon in healthy runners. Soci et al. evaluated the impact of aerobic exercise on miRNA-29c alongside the role of miRNA-29c in cardiac hypertrophy [82]. Female Wistar rats were randomized into three groups (sedentary, training 1 (10 weeks of swimming), training 2 (eight weeks of swimming once a day, one week of running twice a day, and one week of running three times a day)) and exercise was shown to result in inducing cardiac muscle hypertrophy. The training groups showed higher miRNA-29c expression while decreasing collagen (COLIAI and COLIIIAI) gene expression.

Similarly, two studies, Liu et al. and Shi et al., evaluated how miRNA expression following exercise can affect cardiac growth [83,84]. In their study, Liu et al. showed that two exercise models in mice resulted in the upregulation of miRNA-222 and that the inhibition of miRNA-222 was associated with the complete blockade of cardiomyocyte growth [83]. Similarly, Shi et al. found that exercise models tested in mice had an increase in miRNA-17-3p, which promoted cellular proliferation and cardiomyocyte hypertrophy [84]. Unlike skeletal and adipose tissues, no studies have evaluated the association between exercise-induced DNA methylation on cardiac tissue. However, several studies have assessed modifications to histone acetylation by histone deacetylase (HDAC) following exercise. Cox et al. showed that exercise increased the interaction between O-linked β-N-acetylglucosamine transferase and 3A(mSin3A)/HDAC1/HDAC2 complex, which ultimately reversed the process of a diabetic heart [85]. In their study, Lehmann et al. found that exercise-produced N-terminal proteolytically derived fragment of HDAC4 (HDAC4-NT) was associated with preventing heart failure [86].

From a clinical perspective, the impact of miRNA expression on several cardiac diseases such as myocardial infarction, myocardial ischemia/reperfusion (I/R) injury and myocardial fibrosis has been evaluated [80,81,82,83,84]. In addition, miRNA expression following aerobic exercise is associated with cardiac growth and blockade of pathological cardiac remodeling. For example, in Liu et al., following ischemic injury, mice with miRNA-222 upregulation in cardiomyocytes were resistant to dysfunction and adverse cardiac remodeling [83]. Similarly, covalent histone modification has several cardioprotective effects, such as preventing heart failure and diabetic cardiomyopathy [85,86].

## 4. The Combined Effects of Diet and Exercises on Metabolic Diseases through Epigenetic Modifications

As an optimum nutritious diet and exercise are recommended in preventative medicine to protect against several diseases, their combined effects on the epigenetic modulations must be assessed. Understanding the importance of this combined lifestyle intervention in preventative medicine, Hibler et al. [1] conducted the first study to assess the combined impact of diet and exercises through Make Better Choices 2 (MBC2) intervention on epigenome-wide DNA methylation in blood. The study was a 9-month randomized controlled trial, utilizing four distinct lifestyle behaviors, including intake of fruits/vegetables, saturated fats, sedentary leisure screen time and moderate-to-vigorous physical exercises, with methylation inspected at 3 and 9 months. The results suggest that 12 weeks of improved lifestyle choices with regulated dietary and exercise training were associated with changes in DNA methylations at regions of genes linked to tumor suppression, immune cell metabolism and overall aging. Researchers identified over 150 differentially methylated regions at both time points. The consistent finding from the top ten significant regions was that methylation of the region on chromosome 4 was reduced at both time points. This region corresponds to the *interferon regulatory factor 2 (IRF2)* gene, which blocks IRF1-mediated transcriptional activation of interferons alpha and beta. The hypomethylation in this region following the MBC2 intervention could potentially increase the expression of IRF2, which in turn lowers angiogenesis and inflammation. At three months, two regions on chromosome 1 were associated with decreased methylation of the *DUSP5P1* pseudogene in the family of MAPK phosphatases [87]. This gene is described to serve as a tumor suppressor in hematopoietic malignancies. Similar to the cancer-preventative role, other differentially methylated genes critical to metabolic and sensory functions were also determined. Increased methylation of the *PCCA*, an enzyme critical for the proper function of tricarboxylic acid or TCA cycle and energy production in cardiac and skeletal muscle, was observed [1]. In the presence of proinflammatory signals, the TCA cycle is usually upregulated in circulating blood leukocytes; however, studies show that a habitual healthy lifestyle lowers the methylation of proinflammatory genes. Therefore, the hypermethylation of *PCCA* may indicate a decrease in systemic inflammation following the MBC2 intervention, and this association can be further analyzed in future studies. Decreased methylation in *USH1G*, a gene critical to efficient visual and auditory function, is another exciting observation the researchers evaluated [1].

Another commonly studied behavioral pattern is the impact of a high-fat diet and sedentary/exercise training on epigenetic modifications in hepatic cells [88]. In a mice model study, feeding of a high-fat-containing fast-food diet combined with sedentary intervention (FFC) led to lipid accumulation in liver tissues following the stimulation of carnitine palmitoyl-transferase 1a gene transcription, which is a major fatty acid transporter present on the outer membrane of the mitochondrial membrane of liver cells [88]. The diet-induced activation of genes involved in PPAR signaling and fatty acid metabolism, but repressed oxidation reduction genes. Additionally, notable differences were found in the mice group fed a fast-food diet and exercised (FFE). There appeared to be a significant distinction between positive exercise and adverse fast-food diet effects through the activation of fat metabolism and upregulation of genes mediating carbohydrate and lipid metabolism, muscle development and oxidation–reduction processes. This upregulation was unique to the FFE group, as they exhibited more hypomethylation, positively related to transcriptional upregulation. The exercise intervention targeted a significant fraction of uniquely methylated CpG sites in the FFC group. It suppressed nearly half of them, revealing exercise’s strikingly protective effects at the epigenome level. Some of these significant FFE-rescued regions included *Cdh1*, a widely known tumor suppressor and a member of the cytochrome p450 superfamily *Cyp7b1*. Down-regulation via hypermethylation of these two genes in FFC has been related to enhanced cell proliferation, reduced detoxification capacity, and elevated metabolic risk. The antagonistic effects of exercise against abnormal gene regulation depicted here suggests that the beneficial effects of lifestyle modifications through epigenetic reprogramming serve as the molecular link to initiate positive phenotypic alterations [88].

Another study by Wu et al. revealed that DNA methylation is described to be involved in the initiation of non-alcoholic fatty liver disease (NAFLD) that extends from hepatic steatosis to non-alcoholic steatohepatitis (NASH), which can give rise to hepatocellular carcinoma, T2DM and cardiovascular diseases, influenced by dietary and exercise patterns [78]. The study was conducted with a population of healthy and NAFLD-affected individuals aged 50–65 years randomly assigned to four groups—exercise (Ex), low-carbohydrate diet (LCD), exercise plus diet intervention (ELCD) and no intervention (No) groups—for six months. Analyzing the DNA methylation patterns on the blood genome determined that both the LCD and ELCD groups on NAFLD can stimulate the same decreased DNA methylation modifications on promoters of critical genes such as *GAB2* or GRB2-associated binding protein 2, a candidate gene mediating hepatic steatosis and steatohepatitis [78]. It also plays essential roles in differentiation, proliferation, and cell migration by recruiting factors in multiple cells.

This finding was then validated in a subsequent NASH mice model study, where an increase in the DNA methylation level of CpG island within *GAB2* was detected in liver and adipose tissues in low-fat diet (LFD) and exercise plus low-fat diet (ELFD) intervention groups. In terms of finding the optimal lifestyle intervention for NAFLD and NASH through DNA methylation, the results determined that repression of mRNA expression of *GAB2* in NASH mice models after ELFD intervention could provide an effective treatment strategy for NASH. Through this study, researchers demonstrated that DNA methylation differences in the blood genome could mirror epigenetic rewiring in target tissues of critical biological functions such as adipose tissues and the liver. However, as only CpG sites of promoters in the blood genome were included in the human trial for screening differentially methylated gene regions, further studies on DNA methylation in other gene regions of different tissue types may be advantageous [78].

## 5. Conclusions and Perspective

Based on the review, the relationship between exercise interventions and a healthy diet can significantly prevent susceptibility to chronic and metabolic illnesses mediated by epigenetic mechanisms. The increasing evidence showing the transgenerational effects of these lifestyle factors and the irreversible nature of epigenetic modifications strengthens the need for further studies focused on metabolic outcomes.

Understanding the molecular mechanisms mediated by, or resulting in, epigenetic signatures following diet and adaptive responses to exercise may allow the discovery of additional biomarkers to find therapeutic targets to prevent susceptibility to chronic illnesses such as T2DM, obesity, cancer, and cardiovascular risks. Future studies should further define the role of different epigenetic mechanisms, especially DNA methylation, in mediating exercise’s cardioprotective effects.

## Figures and Tables

**Figure 1 biomolecules-13-00318-f001:**
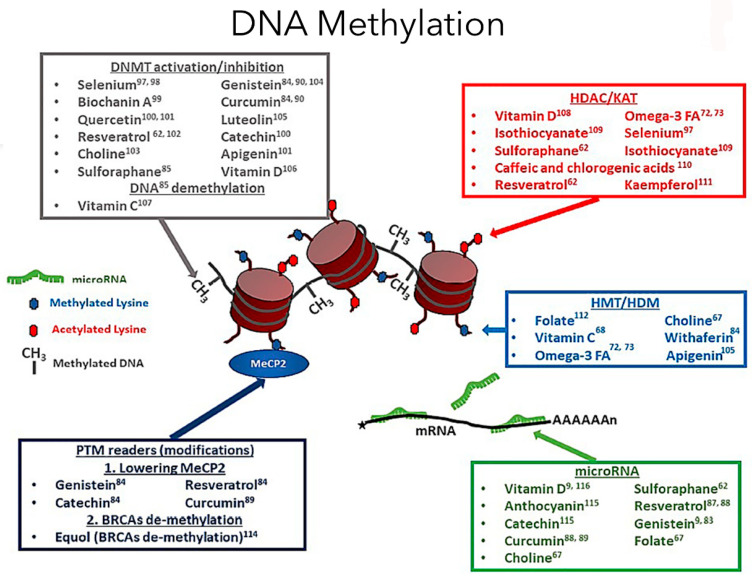
Dietary compounds affect different epigenetic DNA methylation events (adapted from Georgel PT and Georgel P (2021) [30]). DNA methylation is a crucial epigenetic modification involving diet and exercise, involving cytosine pyrimidine position 5′ ring containing a methyl group. In the image, the red cylindrical structure is the histone, and around it is the DNA wound, creating the chromatin structure. Now, the methyl groups will be attached to the DNA regions rich in cytosine-phosphate-guanine (CpG). A methylated gene typically represents an inactive promoter, resulting in chromatin condensation by imparting tight compaction to the structure, preventing transcription [5]. In contrast, the opposite occurs during hypomethylation, leading to gene transcription. The DNA methyltransferase enzymes facilitate the DNA methylation process. Based on the diet factors an individual is exposed to, the genes can be turned on or off based on epigenetic modifications. An epigenome consists of a series of chemical compounds and complex external modifications that can cause changes in the genomic DNA without imparting any changes to a gene’s DNA sequences, giving the cells a unique cellular and developmental identity.

**Figure 2 biomolecules-13-00318-f002:**
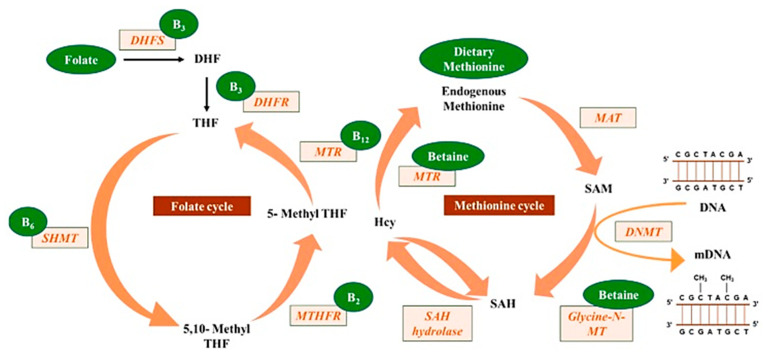
Methyl donors from micronutrients are involved in the one-carbon metabolism, contributing to DNA methylation (reprinted from Mahmoud and Ali (2019) [20]).

**Figure 3 biomolecules-13-00318-f003:**
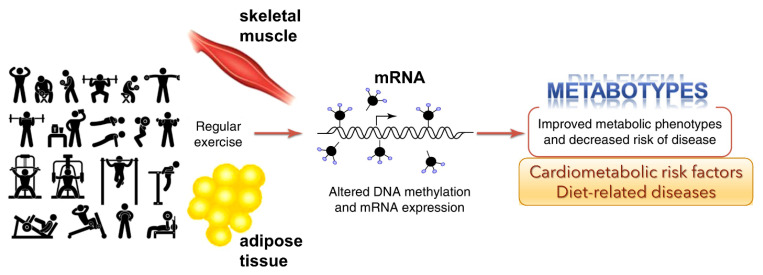
Genome-wide epigenetic modifications in human skeletal muscle and adipose tissue due to regular exercise is linked to DNA methylation and altered mRNA expression (adapted from Ling and Rōnn (2014) [45]). This could potentially improve metabolic phenotypes and decrease the risk of disease.

**Figure 4 biomolecules-13-00318-f004:**
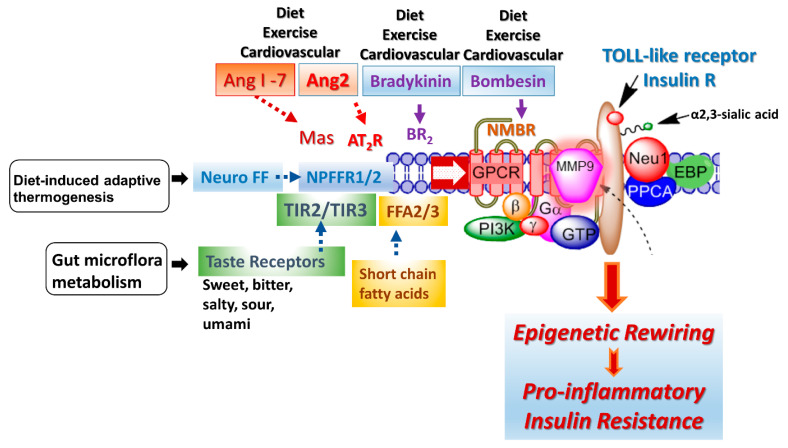
The GPCR receptors, bradykinin (BR2) and angiotensin II receptor type I (AT2R), are tethered with neuromedin B (NMBR), insulin receptor (IRβ) or TOLL-like receptor (TLR), and Neu1 in naïve (unstimulated) and stimulated IR expressing HTC-IR cells (adapted from Haxho et al. (2018) [51]) and TLR expressing live macrophage cell lines and primary macrophage cells [50]. Here, the link regulating the interaction of these molecules and their signaling mechanism(s) on the cell surface reveals a novel biased GPCR signaling process in inducing the IRβ and TLR transactivation signaling, with subsequent activation of Neu1 sialidase and the modification of the receptor glycosylation. Notes: GPCR agonists can potentiate a biased neuromedin B (NMBR)-IR signaling via MMP-9 activation to induce Neu1 sialidase. Activated MMP-9 removes the elastin-binding protein (EBP) as part of β-galactosidase/Neu1 and protective protein cathepsin A (PPCA). Activated Neu1 in turn hydrolyzes α-2,3 sialyl residues of IRβ at the ectodomain to remove steric hindrance, facilitate IRβ subunits association and activate IR tyrosine kinase. Activated phospho-IRβ subunits phosphorylate insulin receptor substrate-1pIRS1, which initiates intracellular insulin signaling via the Ras-MAPK and the PI3K-Akt pathway, promoting epigenetic rewiring and insulin resistance.

## Data Availability

All data needed to evaluate the conclusions in the paper are present in the paper.

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
