# Peer review of "Restoring Epigenetic Reprogramming with Diet and Exercise to Improve Health-Related Metabolic Diseases"

_biomolecules, 2023, doi:10.3390/biom13020318_

Round 1

Reviewer 1 Report

The authors systemically summrized diet and exercise improving metabolic diseases through epigenetic mechanisms. The following problems exist in this paper:

1. There are 12 parts in the thesis, which are a little disorderly. There is no parallel or progressive relationship between each title. The structure of the thesis should be revised as follows:

Introduction

Impact of diets on metabolic diseases through epigenetic modifications

Impact of exercises on metabolic diseases through epigenetic modifications

The combined effects of diets and exercises on metabolic diseases through epigenetic modifications

Conclusion and perspective

2. In TOC diagrams and manuscript, exercise-induced changes in gut microbes, and then epigenetic changes through metabolites, may be only one of the molecular mechanisms, may also affect metabolic diseases in other ways.

3. Besides methylation, there are other epigenetic modulations to ameliorate metabolic diseases

4. Some important literature of the field published in the top journals are not cited. For example:

You are affected by what your parents eat: Diet, epigenetics, transgeneration and intergeneration. Trends in Food Science & Technology, 2020, 100: 248–261.

Dietary polysaccharides exert biological functions via epigenetic regulations: Advance and prospectives. Critical Reviews in Food Science and Nutrition. DOI 10.1080/10408398.2021.1944974

Author Response

Reviewer #1

Comments and Suggestions for Authors

The authors systemically summarized diet and exercise improving metabolic diseases through epigenetic mechanisms. The following problems exist in this paper:

  1. There are 12 parts in the thesis, which are a little disorderly. There is no parallel or progressive relationship between each title. The structure of the thesis should be revised as follows:
  • Introduction
  • Impact of diets on metabolic diseases through epigenetic modifications
  • Impact of exercises on metabolic diseases through epigenetic modifications
  • The combined effects of diets and exercises on metabolic diseases through epigenetic modifications
  • Conclusion and perspective

Author response: Thank you kindly for these comments and suggestions. We have revised the manuscript accordingly.

  1. In TOC diagrams and manuscript, exercise-induced changes in gut microbes, and then epigenetic changes through metabolites, may be only one of the molecular mechanisms, may also affect metabolic diseases in other ways.

Author response: Thank you kindly for these comments. We have revised the manuscript accordingly. We have included a very nice section on “Proposed cellular and biochemical changes causing epigenetic changes.” Line 398-458.

  1. Besides methylation, there are other epigenetic modulations to ameliorate metabolic diseases

Author response: Thank you kindly for these comments. We have included a section on “B vitamins, One Carbon Metabolic Molecules, Biochemical Conduits interlinking DNA methylation, Associated Proteins, and RNA.” Line 214-246. We have also added Figure 2. Methyl donors from micronutrients are involved in the one carbon metabolism, contributing to DNA methylation. Citation: Taken from Mahmoud and Ali [21] Nutrients 2019, 11, 608; doi:10.3390/nu11030608.

  1. Some important literature of the field published in the top journals are not cited. For example:

You are affected by what your parents eat: Diet, epigenetics, transgeneration and intergeneration. Trends in Food Science & Technology, 2020, 100: 248–261.

Dietary polysaccharides exert biological functions via epigenetic regulations: Advance and prospectives. Critical Reviews in Food Science and Nutrition. DOI 10.1080/10408398.2021.1944974

Author response: Thank you kindly for these comments. We have included these articles and cited them in the revised manuscript.

Reviewer 2 Report

The review by Abraham and colleagues adds to an already rich body of literure related to the so called environmental epigenetics. The topic is interesting and offers room for many publications, but the present review appears a little too superficial in the present form to significantly contribute to the field. Moreover, I understand the paper is intended for a special issue on CVD, but it is not really framed on this specific topic.

Specific comments:

1) English style could be improved, in particualr in the abstract where the incipit (for exampel) is not particularly correct ("the concept.....predicts effects")

2) lines 82-83: the idea that DNA methylation is limited to CpG cytosines is old and limited. Many recent reports indicate that non-CpG methylation is high also in somatic and differentiated tissues and the low detection is caused by technical biases

3) lines 89-90 is reducting and misleading. Saying that "DNA methyaltion is positively correlated with gene expression" literally means that when DNA is methylated the gene is expressed...which is exactly the contrary of the methylation-based regulation

4) lines 107-109 the sentence (and the whoe paragraph in general) is not really linked to "inherited" epigenome. It is much more a question of maternal/paternal influence on offspring gene expression, not necessarily involving epigenetic changes in the offspring. Evidence in this sense, in humans, are few. I suggest to revise the whole paragraph and be more cautious.

5) lines 126-127 and related paragraph: the enthusiasm of the authors about the beneficial epigenetic effects of the vegan diet should be revised and discussed on the basis of the well know vegan-associated B12 deficiency, leading to hypomethylation and possible diseased aging. in general, hypomethylated status is associated with accelerated aging and aging-related diseases, so the authors should better substantiate this paragraph. As a matter of fact, they seem to suggest that diet-induced hypomethylation is positive...this is not true.

6) lines 315-322: the authors discuss "two types" of epigenetic effects....but in the aend thay affirm both influence the epigeneome in an aspecific manner. This part need to be better clarified and discussed. The authors should take into account that nutritional modifiers of epigenetics, may act through changes of the biochemical pathways regulating the epigenetic markers (e.g. the B vitamins, modulating the one-carbon metabolsim) and that these changes are likely resulting in gene-specific epigenetic modulation since only the genes "physiologically" modulated by these pathways are affected.

7) all the literature related to the B vitamins in DNA methylation is omitted

Author Response

Reviewer #2

Comments and Suggestions for Authors

The review by Abraham and colleagues adds to an already rich body of literature related to the so called environmental epigenetics. The topic is interesting and offers room for many publications, but the present review appears a little too superficial in the present form to significantly contribute to the field. Moreover, I understand the paper is intended for a special issue on CVD, but it is not really framed on this specific topic.

Author response: Thank you kindly for these comments and we respect your conclusions. However, the review article brings in a novel prospective to the field from different clinical and scientific areas. The question to consider is “What are three risks of CVD? What are the risk factors for cardiovascular disease?” The most important behavioural risk factors of heart disease and stroke are unhealthy diet, physical inactivity, tobacco use and harmful use of alcohol. We have articulated these areas in the review.

Specific comments:

1) English style could be improved, in particular in the abstract where the incipit (for example) is not particularly correct ("the concept.....predicts effects")

Author response: Thank you kindly for these comments. We corrected the first sentence of the abstract. “ABSTRACT: Epigenetic reprogramming predicts the long-term functional health effects of health-related metabolic disease. This epigenetic reprogramming is activated by exogenous or endogenous insults, leading to altered healthy and different disease states.”

Also, we routinely use Grammarly software to detect and correct areas for sentence structure, for correctness, clarity, engagement and delivery.

2) lines 82-83: the idea that DNA methylation is limited to CpG cytosines is old and limited. Many recent reports indicate that non-CpG methylation is high also in somatic and differentiated tissues and the low detection is caused by technical biases

Author response: Thank you kindly for these comments. We have included a section on this issue. For example, “[6]. Recent reports have presented that non-CpG methylation is upregulated in differentiated and somatic tissues, and the low detection is caused by technical biases [11]. If position 5 of the CpG island is methylated, it represents an inactive promoter resulting in chromatin condensation by imparting tight compaction to the structure, preventing transcription [6]. Thus, DNA methylation plays a significant role in tissue-specific gene regulation and transcription throughout life. It has been suggested that DNA methylation is a natural link between genetic susceptibility and environmental exposures in common dis-eases [12]. This methylation process is facilitated by the enzyme DNA methyltransferases (DNMTs) [6].In recent years, reports have shown that a significant fraction of DNA meth-ylation sites positively correlate with gene expression [13], challenging the traditional view that DNA methylation represses gene expression.” Lines 84-94

3) lines 89-90 is reducting and misleading. Saying that "DNA methylation is positively correlated with gene expression" literally means that when DNA is methylated the gene is expressed...which is exactly the contrary of the methylation-based regulation

Author response: Thank you kindly for these comments. We have included a section on this issue and made the correction. For example, “In recent years, reports have shown that a significant fraction of DNA methylation sites positively correlate with gene expression [13], challenging the traditional view that DNA methylation represses gene expression. However, Wan et al. [14] demonstrated that the expression of genes negatively correlated tissue-specific differentially methylated regions (T-DMRs) are enriched for functions carried out in adult tissues. In contrast, the positively correlated genes were enriched for negative regulators such as transcriptional repressors. Interestingly, this two-tier mechanism regulated by positive T-DMRs may be specific to the development and the establishment of tissue-specific expression. Also, gene regulation can be modulated depending on the location of methylation in the transcriptional site [15].”

4) lines 107-109 the sentence (and the whoe paragraph in general) is not really linked to "inherited" epigenome. It is much more a question of maternal/paternal influence on offspring gene expression, not necessarily involving epigenetic changes in the offspring. Evidence in this sense, in humans, are few. I suggest to revise the whole paragraph and be more cautious.

Author response: Thank you kindly for these comments. We have included a section on this issue and made the correction. For example, “However, Wan et al. [14] demonstrated that the expression of genes negatively correlated tissue-specific differentially methylated regions (T-DMRs) are enriched for functions carried out in adult tissues. In contrast, the positively correlated genes were enriched for negative regulators such as transcriptional repressors. Interestingly, this two-tier mechanism regulated by positive T-DMRs may be specific to the development and the establishment of tissue-specific expression. Also, gene regulation can be modulated depending on the lo-cation of methylation in the transcriptional site [15].

If positive T-DMRs are specific to development [14], the question is whether epigenetics and epigenetic inheritance influence evolution [16]. Trans-generational epigenetic in-heritance must be considered when estimating quantitative genetic parameters, but it can also respond to selection and influence adaptation to new environments [14]. For example, the transgenerational effects of poor maternal diet during pregnancy can lead to health disorders and disease susceptibility or cancer throughout the offspring’s life [17,18].”

5) lines 126-127 and related paragraph: the enthusiasm of the authors about the beneficial epigenetic effects of the vegan diet should be revised and discussed on the basis of the well know vegan-associated B12 deficiency, leading to hypomethylation and possible diseased aging. in general, hypomethylated status is associated with accelerated aging and aging-related diseases, so the authors should better substantiate this paragraph. As a matter of fact, they seem to suggest that diet-induced hypomethylation is positive...this is not true.

Author response: Thank you kindly for these comments. We have included a section on this issue and made the correction. For example, “The plant-based or ‘vegan’ diet generally contains whole grains, legumes, vegetables, fruits and nuts and abstention from meat and dairy products. They are rich in polyphenols and secondary plant metabolites that could inhibit DNA methyltransferase (DNMT) from preventing cancer, metabolic diseases, and other chronic illnesses. Contrarily, the vegan diet has also been associated with vegan-associated B12 deficiency, possibly leading to diseased aging and hypomethylation [26]. DNA methylation, catalyzed by DNMTs, is essential in maintaining genome stability (Figure 1). DNMT’s aberrant expression and its disruption of DNA methylation patterns are closely associated with many forms of cancer [27].”

6) lines 315-322: the authors discuss "two types" of epigenetic effects....but in the and they affirm both influence the epigenome in an a specific manner. This part needs to be better clarified and discussed. The authors should take into account that nutritional modifiers of epigenetics, may act through changes of the biochemical pathways regulating the epigenetic markers (e.g. the B vitamins, modulating the one-carbon metabolism) and that these changes are likely resulting in gene-specific epigenetic modulation since only the genes "physiologically" modulated by these pathways are affected.

Author response: Thank you kindly for these comments. We have included a section on this issue and made the correction. For example, we have included a whole section on “B vitamins, One Carbon Metabolic Molecules, Biochemical Conduits interlinking DNA methylation, Associated Proteins, and RNA” Lines 214-264 with a new Figure 2. Methyl donors from micronutrients are involved in the one carbon metabolism, contributing to DNA methylation. Citation: Taken from Mahmoud and Ali [21] Nutrients 2019, 11, 608; doi:10.3390/nu11030608. Copyright © 2019 Mahmoud and Ali. This open-access article is distributed under the terms of the Creative Commons Attribution License (CC BY). The use, distribution or reproduction in other forums is permitted, provided the original author(s) and the copyright owner(s) are credited. The original publication in this journal is cited, per accepted academic practice. No use, distribution or reproduction is permitted, which violates these terms.

7) all the literature related to the B vitamins in DNA methylation is omitted

Author response: Thank you kindly for these comments. We have included a section on this issue and made the correction. Please see above under item 6.

Round 2

Reviewer 2 Report

The authors made some adjustement based on my previous comments.